# DisARM: An Antithetic Gradient Estimator for Binary Latent Variables

**Zhe Dong**
Google Research, Brain Team
zhedong@google.com

**Andriy Mnih**
DeepMind
amnih@google.com

**George Tucker**
Google Research, Brain Team
gjt@google.com

## Abstract

Training models with discrete latent variables is challenging due to the difficulty of estimating the gradients accurately. Much of the recent progress has been achieved by taking advantage of continuous relaxations of the system, which are not always available or even possible. The Augment-REINFORCE-Merge (ARM) estimator provides an alternative that, instead of relaxation, uses continuous augmentation. Applying antithetic sampling over the augmenting variables yields a relatively low-variance and unbiased estimator applicable to any model with binary latent variables. However, while antithetic sampling reduces variance, the augmentation process increases variance. We show that ARM can be improved by analytically integrating out the randomness introduced by the augmentation process, guaranteeing substantial variance reduction. Our estimator, *DisARM*, is simple to implement and has the same computational cost as ARM. We evaluate DisARM on several generative modeling benchmarks and show that it consistently outperforms ARM and a strong independent sample baseline in terms of both variance and log-likelihood. Furthermore, we propose a local version of DisARM designed for optimizing the multi-sample variational bound, and show that it outperforms VIMCO, the current state-of-the-art method.

## 1 Introduction

We often require the gradient of an expectation with respect to the parameters of the distribution. In all but the simplest settings, the expectation is analytically intractable and the gradient is estimated using Monte Carlo sampling. This problem is encountered, for example, in modern variational inference, where we would like to maximize a variational lower bound with respect to the parameters of the variational posterior. The pathwise gradient estimator, also known as the reparameterization trick, comes close to this ideal and has been instrumental to the success of variational autoencoders (Kingma and Welling, 2014; Rezende et al., 2014). Unfortunately, it can only be used with continuous random variables, and finding a similarly effective estimator for discrete random variables remains an important open problem.

Score-function estimators (Glynn, 1990; Fu, 2006), also known as REINFORCE (Williams, 1992), have historically been the estimators of choice for models with discrete random variables due to their unbiasedness and few requirements. As they usually exhibit high variance, previous work has augmented them with variance reduction methods to improve their practicality (Williams, 1992; Ranganath et al., 2014; Mnih and Gregor, 2014). Motivated by the efficiency of the pathwise estimator, recent progress in gradient estimators for discrete variables has primarily been driven by leveraging gradient information. The original system may only be defined for discrete inputs and hence gradients w.r.t. the random variables may not be defined. If we can construct a continuous relaxation of the

---

Code and additional information: https://sites.google.com/view/disarm-estimator.

system, then we can compute gradients of the continuous system and use them in an estimator (Gu et al., 2016; Jang et al., 2017; Maddison et al., 2017; Tucker et al., 2017; Grathwohl et al., 2018).

While such relaxation techniques are appealing because they result in low variance estimators by taking advantage of gradient information, they are not always applicable. In some cases, the function we compute the expectation of will not be differentiable w.r.t. the random variables, e.g. if it is a table indexed by the variables. In other cases, the computational cost of evaluating the function at the relaxed variable values will be prohibitive, e.g. in conditional computation (Bengio et al., 2013), where discrete variables specify which parts of a large model should be evaluated and using a relaxation would require evaluating the entire model every time.

The recently introduced Augment-REINFORCE-Merge (ARM) estimator (Yin and Zhou, 2019) provides a promising alternative to relaxation-based estimators for binary latent variables. Instead of relaxing the variables, ARM reparameterizes them as deterministic transformations of the underlying continuous variables. Applying antithetic sampling to the REINFORCE estimator w.r.t. the parameters of the underlying continuous distribution yields a highly competitive estimator. We observe that the continuous augmentation, which is the first step in ARM, increases the variance of the REINFORCE estimator, and antithetic sampling is the only reason ARM outperforms REINFORCE on the original binary distribution. We improve on ARM by integrating over the augmenting variables, thus eliminating the unnecessary randomness introduced by the augmentation and reducing the variance of the estimator substantially. We show that the resulting estimator, DisARM, consistently outperforms ARM and is highly competitive with RELAX. Concurrently, Yin et al. (2020) discovered the same estimator, calling it the U2G estimator and demonstrating promising performance on best subset selection tasks. We also derive a version of DisARM for the multi-sample variational bound and show that it outperforms the current state-of-the-art gradient estimator for that objective.

## 2  Background

We consider the problem of optimizing

$$\mathbb{E}_{q_\theta(\mathbf{b})}\left[f_\theta(\mathbf{b})\right],\tag{1}$$

w.r.t. the parameters $\theta$ of a factorial Bernoulli distribution $q_\theta(\mathbf{b})$. This situation covers many problems with discrete latent variables, for example, in variational inference $f_\theta(\mathbf{b})$ could be the instantaneous ELBO (Jordan et al., 1999) and $q_\theta(\mathbf{b})$ the variational posterior.

The gradient with respect to $\theta$ is

$$\nabla_\theta \mathbb{E}_{q_\theta(\mathbf{b})}\left[f_\theta(\mathbf{b})\right] = \mathbb{E}_{q_\theta(\mathbf{b})}\left[f_\theta(\mathbf{b})\nabla_\theta \log q_\theta(\mathbf{b}) + \nabla_\theta f_\theta(\mathbf{b})\right].\tag{2}$$

The second term can typically be estimated with a single Monte Carlo sample, so for notational clarity, we omit the dependence of $f$ on $\theta$ in the following sections. Monte Carlo estimates of the first term can have large variance. Low-variance, unbiased estimators of the first term will be our focus.

### 2.1  Augment-REINFORCE-Merge (ARM)

For exposition, we review the single variable case, and it is straightforward to extend the results to the multi-dimensional setting. Yin and Zhou (2019) use an *antithetically* coupled pair of samples to derive the ARM estimator. By carefully introducing statistical dependency between samples, we can reduce the variance of the estimator over using independent samples without introducing bias because of the linearity of expectations. Intuitively, antithetic sampling chooses "opposite" pairs of samples which ensures that the space is better covered than with independent samples (see (Owen, 2013) for a detailed review). Antithetic sampling reduces the variance of a Monte Carlo estimate if the integrand evaluated at the two samples has negative covariance. While we have no control over $f$, we can exploit properties of the score function $\nabla_\theta \log q_\theta(b)$. Buesing et al. (2016) show that for "location-scale" distributions, antithetically coupled samples have perfectly negatively correlated score functions, which suggests that using antithetic samples to estimate the gradient will be favorable. Unfortunately, the Bernoulli distribution is not a location-scale distribution, so this result is not immediately applicable.

However, the Bernoulli distribution can be reparameterized in terms of the Logistic distribution which is a location-scale distribution. In other words, let $\alpha_\theta$ be the logits of the Bernoulli distribution (which

may be the output of a function parameterized by $\theta$) and $z \sim \text{Logistic}(\alpha_\theta, 1)$, then $b = \mathbb{1}_{z>0} \sim \text{Bernoulli}(\sigma(\alpha_\theta))$, where $\sigma(x)$ is the Logistic function. We also have

$$\mathbb{E}_{q_\theta(b)}\left[f(b)\nabla_\theta \log q_\theta(b)\right] = \nabla_\theta \mathbb{E}_{q_\theta(b)}\left[f(b)\right] = \nabla_\theta \mathbb{E}_{q_\theta(z)}\left[f(\mathbb{1}_{z>0})\right] = \mathbb{E}_{q_\theta(z)}\left[f(\mathbb{1}_{z>0})\nabla_\theta \log q_\theta(z)\right].$$

For Logistic random variables, a natural antithetic coupling is defined by drawing $\epsilon \sim \text{Logistic}(0, 1)$, then setting $z = \epsilon + \alpha_\theta$ and $\tilde{z} = -\epsilon + \alpha_\theta$ such that both $z$ and $\tilde{z}$ have the same marginal distribution, however, they are not independent. With the antithetically coupled pair $(z, \tilde{z})$, we can form the estimator

$$
\begin{aligned}
g_{\text{ARM}}(z, \tilde{z}) &= \tfrac{1}{2}\left(f(\mathbb{1}_{z>0})\nabla_\theta \log q_\theta(z) + f(\mathbb{1}_{\tilde{z}>0})\nabla_\theta \log q_\theta(\tilde{z})\right) \\
&= \tfrac{1}{2}(f(\mathbb{1}_{z>0}) - f(\mathbb{1}_{\tilde{z}>0}))\nabla_\theta \log q_\theta(z) \\
&= \tfrac{1}{2}(f(\mathbb{1}_{1-u<\sigma(\alpha_\theta)}) - f(\mathbb{1}_{u<\sigma(\alpha_\theta)}))\left(2u-1\right)\nabla_\theta \alpha_\theta, \quad\quad (3)
\end{aligned}
$$

where $u = \sigma(z - \alpha_\theta)$ and we use the fact that $\nabla_\theta \log q_\theta(z) = -\nabla_\theta \log q_\theta(\tilde{z})$ (Buesing et al., 2016) because the Logistic distribution is a location-scale distribution. This is the ARM estimator (Yin and Zhou, 2019). Notably, ARM only evaluates $f$ at discrete values, so does not require a continuous relaxation. ARM is unbiased and we expect it to have low variance because the learning signal is a difference of evaluations of $f$. Yin and Zhou (2019) empirically show that it performs comparably or outperforms previous methods. In the scalar setting, ARM is not useful because the exact gradient can be computed with 2 function evaluations, however, ARM can naturally be extended to the multi-dimensional setting with only 2 function evaluations

$$\tfrac{1}{2}(f(\mathbf{b}) - f(\tilde{\mathbf{b}}))\left(2\mathbf{u}-1\right)\nabla_\theta \alpha_\theta, \quad\quad (4)$$

whereas the exact gradient requires exponentially many function evaluations.

## 2.2 Multi-sample variational bounds

Objectives of the form Eq. 1 are often used in variational inference for discrete latent variable models. For example, to fit the parameters of a discrete latent variable model $p_\theta(x, \mathbf{b})$, we can lower bound the log marginal likelihood $\log p_\theta(x) \geq \mathbb{E}_{q_\theta(\mathbf{b}|x)}\left[\log p_\theta(x, \mathbf{b}) - \log q_\theta(\mathbf{b}|x)\right]$, where $q_\theta(\mathbf{b}|x)$ is a variational distribution. Burda et al. (2016) introduced an improved multi-sample variational bound that reduces to the ELBO when $K = 1$ and converges to the log marginal likelihood as $K \to \infty$

$$\mathcal{L} := \mathbb{E}_{\prod_k q_\theta(\mathbf{b}^k|x)}\left[\log \tfrac{1}{K}\sum_k w(\mathbf{b}^k)\right],$$

where $w(\mathbf{b}) = \frac{p(\mathbf{b}, x)}{q(\mathbf{b}|x)}$. We omit the dependence of $w$ on $\theta$ because it is straightforward to account for.

In this case, Mnih and Rezende (2016) introduced a gradient estimator, VIMCO, that uses specialized control variates that take advantage of the structure of the objective

$$\sum_k \left(\log \tfrac{1}{K}\sum_j w(\mathbf{b}^j) - \log \tfrac{1}{K-1}\sum_{j\neq k} w(\mathbf{b}^j)\right)\nabla_\theta \log q_\theta(\mathbf{b}^k|x),$$

which is unbiased because $\mathbb{E}_{\prod_k q_\theta(\mathbf{b}^k|x)}\left[\left(\log \tfrac{1}{K-1}\sum_{j\neq k} w(\mathbf{b}^j)\right)\nabla_\theta \log q_\theta(\mathbf{b}^k|x)\right] = 0$.

## 3 DisARM

Requiring a reparameterization in terms of a continuous variable seems unnatural when the objective (Eq. 1) only depends on the discrete variable. The cost of this reparameterization is an increase in variance. In fact, the variance of $f(\mathbb{1}_{z>0})\nabla_\theta \log q_\theta(z)$ is at least as large as the variance of $f(b)\nabla_\theta \log q_\theta(b)$ because

$$f(b)\nabla_\theta \log q_\theta(b) = \mathbb{E}_{q_\theta(z|b)}\left[f(\mathbb{1}_{z>0})\nabla_\theta \log q_\theta(z)\right], \quad\quad (5)$$

hence

$$\text{Var}(f(\mathbb{1}_{z>0})\nabla_\theta \log q_\theta(z)) = \text{Var}(f(b)\nabla_\theta \log q_\theta(b)) + \mathbb{E}_b\left[\text{Var}_{z|b}(f(\mathbb{1}_{z>0})\nabla_\theta \log q_\theta(z))\right],$$

i.e., an instance of conditioning (Owen, 2013). So, while ARM reduces variance via antithetic coupling, it also increases variance due to the reparameterization. It is not clear that this translates to an overall reduction in variance. In fact, as we show empirically, a two-independent-samples

REINFORCE estimator with a leave-one-out baseline performs comparably or outperforms the ARM estimator (e.g., Table 1).

The relationship in Eq. 5 suggests that it might be possible to perform a similar operation on the ARM estimator. Indeed, the key insight is to simultaneously condition on the pair $(b, \tilde{b}) = (\mathbb{1}_{z>0}, \mathbb{1}_{\tilde{z}>0})$. First, we derive the result for scalar $b$, then extend it to the multi-dimensional setting. Integrating out $z$ conditional on $(b, \tilde{b})$, results in our proposed estimator

$$
\begin{aligned}
g_{\text{DisARM}}(b, \tilde{b}) &:= \mathbb{E}_{q(z|b,\tilde{b})} [g_{\text{ARM}}] = \tfrac{1}{2} \mathbb{E}_{q(z|b,\tilde{b})} \left[ (f(\mathbb{1}_{z>0}) - f(\mathbb{1}_{\tilde{z}>0})) \nabla_\theta \log q_\theta(z) \right] \\
&= \tfrac{1}{2}(f(b) - f(\tilde{b})) \mathbb{E}_{q(z|b,\tilde{b})} \left[ \nabla_\theta \log q_\theta(z) \right] \\
&= \tfrac{1}{2}(f(b) - f(\tilde{b})) \left( (-1)^{\tilde{b}} \mathbb{1}_{b \neq \tilde{b}} \sigma(|\alpha_\theta|) \right) \nabla_\theta \alpha_\theta.
\end{aligned}
\tag{6}
$$

See Appendix A for a detailed derivation. Note that $\mathbb{E}_{q(z|b,\tilde{b})} [\nabla_\theta \log q_\theta(z)]$ vanishes when $b = \tilde{b}$. While this does not matter for the scalar case, it will prove useful for the multi-dimensional case. We call the estimator DisARM because it integrates out the continuous randomness in ARM and only retains the *discrete* component. Similarly to above, we have that the variance of DisARM is upper bounded by the variance of ARM

$$
\text{Var}(g_{\text{ARM}}) = \text{Var}(g_{\text{DisARM}}) + \mathbb{E}_{b,\tilde{b}} \left[ \text{Var}_{z|b,\tilde{b}}(g_{\text{ARM}}) \right] \geq \text{Var}(g_{\text{DisARM}}).
$$

## 3.1 Multi-dimensional case

Now, consider the case where $\mathbf{b}$ is multi-dimensional. Although the distribution is factorial, $f$ may be a complex nonlinear function. Focusing on a single dimension of $\alpha_\theta$, we have

$$
\begin{aligned}
\nabla_{(\alpha_\theta)_i} \mathbb{E}_{q_\theta(\mathbf{b})} [f(\mathbf{b})] &= \nabla_{(\alpha_\theta)_i} \mathbb{E}_{\mathbf{b}_i} \left[ \mathbb{E}_{\mathbf{b}_{-i}} [f(\mathbf{b}_{-i}, \mathbf{b}_i)] \right] \\
&= \mathbb{E}_{\mathbf{b}_i, \tilde{\mathbf{b}}_i} \left[ \tfrac{1}{2} \left( \mathbb{E}_{\mathbf{b}_{-i}} [f(\mathbf{b}_{-i}, \mathbf{b}_i)] - \mathbb{E}_{\mathbf{b}_{-i}} [f(\mathbf{b}_{-i}, \tilde{\mathbf{b}}_i)] \right) \left( (-1)^{\tilde{\mathbf{b}}_i} \mathbb{1}_{\mathbf{b}_i \neq \tilde{\mathbf{b}}_i} \sigma(|(\alpha_\theta)_i|) \right) \right],
\end{aligned}
$$

which follows from applying Eq. 6 where the function is now $\mathbb{E}_{\mathbf{b}_{-i}} [f(\mathbf{b}_{-i}, \mathbf{b}_i)]$, and $\mathbf{b}_{-i}$ denotes the vector of samples obtained by leaving out $i$th dimension. Then, because expectations are linear, we can couple the inner expectations

$$
\begin{aligned}
\mathbb{E}_{\mathbf{b}_i, \tilde{\mathbf{b}}_i} & \left[ \tfrac{1}{2} (\mathbb{E}_{\mathbf{b}_{-i}} [f(\mathbf{b}_{-i}, \mathbf{b}_i)] - \mathbb{E}_{\mathbf{b}_{-i}} [f(\mathbf{b}_{-i}, \tilde{\mathbf{b}}_i)]) \left( (-1)^{\tilde{\mathbf{b}}_i} \mathbb{1}_{\mathbf{b}_i \neq \tilde{\mathbf{b}}_i} \sigma(|(\alpha_\theta)_i|) \right) \right] \\
&= \mathbb{E}_{\mathbf{b}_i, \tilde{\mathbf{b}}_i} \left[ \tfrac{1}{2} (\mathbb{E}_{\mathbf{b}_{-i}, \mathbf{b}'_{-i}} [f(\mathbf{b}_{-i}, \mathbf{b}_i) - f(\mathbf{b}'_{-i}, \tilde{\mathbf{b}}_i)]) \left( (-1)^{\tilde{\mathbf{b}}_i} \mathbb{1}_{\mathbf{b}_i \neq \tilde{\mathbf{b}}_i} \sigma(|(\alpha_\theta)_i|) \right) \right],
\end{aligned}
$$

where we are free to choose any joint distribution on $(\mathbf{b}_{-i}, \mathbf{b}'_{-i})$ that maintains the marginal distributions. A natural choice satisfying this constraint is to draw $(\mathbf{b}, \tilde{\mathbf{b}})$ as an antithetic pair (independently for each dimension), then we can form the multi-dimensional DisARM estimator of $\nabla_{(\alpha_\theta)_i}$

$$
\tfrac{1}{2}(f(\mathbf{b}) - f(\tilde{\mathbf{b}})) \left( (-1)^{\tilde{\mathbf{b}}_i} \mathbb{1}_{\mathbf{b}_i \neq \tilde{\mathbf{b}}_i} \sigma(|(\alpha_\theta)_i|) \right).
\tag{7}
$$

Notably, whenever $\mathbf{b}_i = \tilde{\mathbf{b}}_i$, the gradient estimator vanishes exactly. In contrast, the multi-dimensional ARM estimator of $\nabla_{(\alpha_\theta)_i}$ (Eq. 4) vanishes only when $\mathbf{b} = \tilde{\mathbf{b}}$ in all dimensions, which occurs seldomly when $\mathbf{b}$ is high dimensional. The estimator for $\nabla_\theta$ is obtained by summing over $i$:

$$
g_{\text{DisARM}}(\mathbf{b}, \tilde{\mathbf{b}}) = \sum_i \left( \tfrac{1}{2}(f(\mathbf{b}) - f(\tilde{\mathbf{b}})) \left( (-1)^{\tilde{\mathbf{b}}_i} \mathbb{1}_{\mathbf{b}_i \neq \tilde{\mathbf{b}}_i} \sigma(|(\alpha_\theta)_i|) \right) \nabla_\theta (\alpha_\theta)_i \right).
\tag{8}
$$

## 3.2 Extension to multi-sample variational bounds

We could naïvely apply DisARM to the multi-sample objective, however, our preliminary experiments did not suggest this improved performance over VIMCO. However, we can obtain an estimator similar to VIMCO (Mnih and Rezende, 2016) by applying DisARM to the multi-sample objective *locally*, once for each sample. Recall that in this setting, our objective is the multi-sample variational lower bound (Burda et al., 2016)

$$
\mathcal{L} := \mathbb{E}_{\prod_k q_\theta(\mathbf{b}^k)} \left[ \log \frac{1}{K} \sum_k w(\mathbf{b}^k) \right] = \mathbb{E}_{\prod_k q_{\theta^k}(\mathbf{b}^k)} \left[ \log \frac{1}{K} \sum_k w(\mathbf{b}^k) \right],
$$

where to simplify notation, we introduced dummy variables $\theta^k = \theta$, so that $\nabla_\theta \mathcal{L} = \sum_k \frac{\partial \mathcal{L}}{\partial \theta^k}$. Now, let $f_{\mathbf{b}^{-k}}(\mathbf{d}) = \log \frac{1}{K} \left( \sum_{\mathbf{c} \in \mathbf{b}^{-k}} w(\mathbf{c}) + w(\mathbf{d}) \right)$ with $\mathbf{b}^{-k} := (\mathbf{b}^1, \ldots, \mathbf{b}^{k-1}, \mathbf{b}^{k+1}, \ldots, \mathbf{b}^K)$, so that

$$\frac{\partial \mathcal{L}}{\partial \theta^k} = \frac{\partial \mathcal{L}}{\partial \theta^k} \mathbb{E}_{\mathbf{b}^k} \left[ \mathbb{E}_{\mathbf{b}^{-k}} \left[ f_{\mathbf{b}^{-k}}(\mathbf{b}^k) \right] \right] = \frac{\partial \mathbb{E}_{\mathbf{b}^k} \left[ \mathbb{E}_{\mathbf{b}^{-k}} \left[ f_{\mathbf{b}^{-k}}(\mathbf{b}^k) \right] \right]}{\partial \alpha_{\theta^k}} \frac{\partial \alpha_{\theta^k}}{\partial \theta^k}.$$

Then by applying Eq. 7 to $\mathbb{E}_{\mathbf{b}^{-k}} \left[ f_{\mathbf{b}^{-k}} \right]$, we have that $\left( \frac{\partial \mathcal{L}}{\partial \alpha_{\theta^k}} \mathbb{E}_{\mathbf{b}^k} \left[ \mathbb{E}_{\mathbf{b}^{-k}} \left[ f_{\mathbf{b}^{-k}}(\mathbf{b}^k) \right] \right] \right)_i$ is

$$\mathbb{E}_{\mathbf{b}^k, \tilde{\mathbf{b}}^k} \left[ \frac{1}{2} \left( \mathbb{E}_{\mathbf{b}^{-k}} \left[ f_{\mathbf{b}^{-k}}(\mathbf{b}^k) \right] - \mathbb{E}_{\mathbf{b}^{-k}} \left[ f_{\mathbf{b}^{-k}}(\tilde{\mathbf{b}}^k) \right] \right) \left( \mathbb{1}_{\mathbf{b}_i^k \neq \tilde{\mathbf{b}}_i^k} (-1)^{\tilde{\mathbf{b}}_i^k} \sigma(|(\alpha_{\theta^k})_i|) \right) \right].$$

We can form an unbiased estimator by drawing $K$ antithetic pairs $\mathbf{b}^1, \tilde{\mathbf{b}}^1, \ldots, \mathbf{b}^K, \tilde{\mathbf{b}}^K$ and forming

$$\frac{1}{4} \left( f_{\mathbf{b}^{-k}}(\mathbf{b}^k) - f_{\mathbf{b}^{-k}}(\tilde{\mathbf{b}}^k) + f_{\tilde{\mathbf{b}}^{-k}}(\mathbf{b}^k) - f_{\tilde{\mathbf{b}}^{-k}}(\tilde{\mathbf{b}}^k) \right) \left( \mathbb{1}_{\mathbf{b}_i^k \neq \tilde{\mathbf{b}}_i^k} (-1)^{\tilde{\mathbf{b}}_i^k} \sigma(|(\alpha_\theta)_i|) \right), \quad (9)$$

for the gradient of the $i$th dimension and $k$th sample. Conveniently, we can compute $w(\mathbf{b}^1), w(\tilde{\mathbf{b}}^1), \ldots, w(\mathbf{b}^K), w(\tilde{\mathbf{b}}^K)$ once and then compute the estimator for all $k$ and $i$ without additional evaluations of $w$. As a result, the computation associated with this estimator is the same as for VIMCO with $2K$ samples, and thus we use it as a baseline comparison in our experiments. We could average over further configurations to reduce the variance of our estimate of $\mathbb{E}_{\mathbf{b}^{-k}}[f_{\mathbf{b}^{-k}}]$, however, we leave evaluating this to future work.

## 4   Related Work

Virtually all unbiased gradient estimators for discrete variables in machine learning are variants of the score function (SF) estimator (Fu, 2006), also known as REINFORCE or the likelihood-ratio estimator. As the naive SF estimator tends to have high variance, these estimators differ in the variance reduction techniques they employ. The most widely used of these techniques are control variates (Owen, 2013). Constant multiples of the score function itself are the most widely used control variates, known as baselines.[1] The original formulation of REINFORCE (Williams, 1992) already included a baseline, as did its earliest specializations to variational inference (Paisley et al., 2012; Wingate and Weber, 2013; Ranganath et al., 2014; Mnih and Gregor, 2014). When the function $f(\mathbf{b})$ is differentiable, more sophisticated control variates can be obtained by incorporating the gradient of $f$. MuProp (Gu et al., 2016) takes the "mean field" approach by evaluating the gradient at the means of the latent variables, while REBAR (Tucker et al., 2017) obtains the gradient by applying the Gumbel-Softmax / Concrete relaxation (Jang et al., 2017; Maddison et al., 2017) to the latent variables and then using the reparameterization trick. RELAX (Grathwohl et al., 2018) extends REBAR by augmenting it with a free-form control variate. While in principle RELAX is generic because the free-form control variate can be learned from scratch, the strong performance previously reported (and in this paper) relies on a continuous relaxation of the discrete function and only learns a small deviation from this hard-coded relaxation.

The ARM (Yin and Zhou, 2019) estimator uses antithetic sampling to reduce the variance of the underlying score-function estimator applied to the Logistic augmentation of Bernoulli variables. Antithetic sampling has also been recently used to reduce the gradient variance for the reparameterization trick (Ren et al., 2019; Wu et al., 2019). The general approach behind the ARM estimator has been generalized to the categorical case by Yin et al. (2019).

Computing the expectation w.r.t. some of the random variables analytically is another powerful variance reduction technique, known as conditioning or Rao-Blackwellization (Owen, 2013). This is the technique we apply to ARM to obtain DisARM. Local Expectation Gradients (Titsias and Lázaro-Gredilla, 2015) apply this idea to one latent variable at a time, computing its conditional expectation given the state of the remaining variables in a sample from the variational posterior.

## 5   Experimental Results

Our goal was variance reduction to improve optimization, so we compare DisARM to the state-of-the-art methods: ARM (Yin and Zhou, 2019) and RELAX (Grathwohl et al., 2018) for the general case

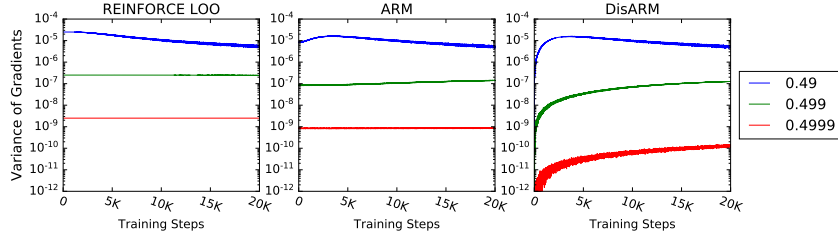

Figure 1: Variance of the gradient estimators for the toy problem (Section 5.1). The variance was computed using 5000 Monte Carlo samples.

and VIMCO (Mnih and Rezende, 2016) for the multi-sample variational bound. As we mentioned before, ARM and DisARM are more generally applicable than RELAX, however, we include it for comparison. We also include a two-independent-sample REINFORCE estimator with a leave-one-out baseline (REINFORCE LOO, Kool et al., 2019). This is a simple, but competitive method that has been omitted from previous works. First, we evaluate our proposed gradient estimator, DisARM, on an illustrative problem, where we can compute exact gradients. Then, we train a variational auto-encoder (Kingma and Welling, 2014; Rezende et al., 2014) (VAE) with Bernoulli latent variables with the ELBO and the multi-sample variational bound on three generative modeling benchmark datasets.

## 5.1 Learning a Toy Model

We start with a simple illustrative problem, introduced by Tucker et al. (2017), where the goal is to maximize $\mathbb{E}_{b\sim\text{Bernoulli}(\sigma(\phi))}\left[(b-p_0)^2\right]$. We apply DisARM to the three versions of this task ($p_0 \in \{0.49, 0.499, 0.4999\}$), and compare its performance to ARM and REINFORCE LOO in Figure 1[2], with full comparison in Appendix Figure 5. DisARM exhibits lower variance than REINFORCE LOO and ARM, especially for the more difficult versions of the problem as $p_0$ approaches 0.5.

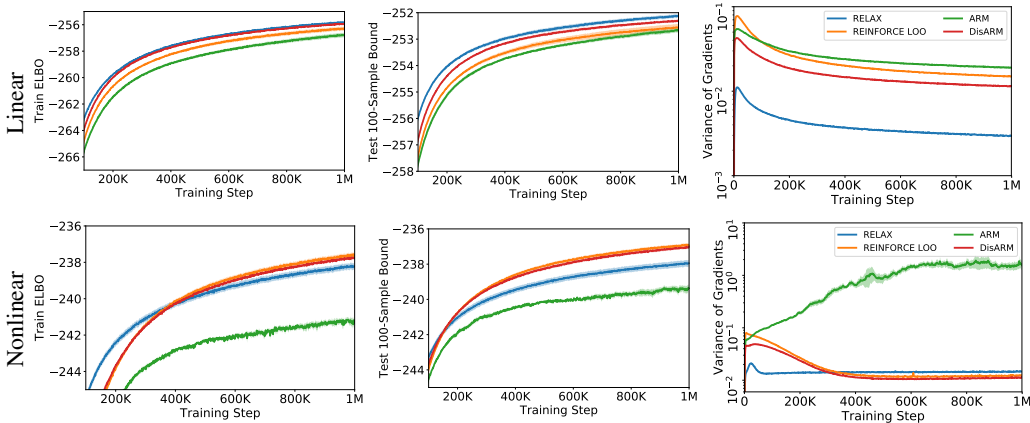

Figure 2: Training a Bernoulli VAE on FashionMNIST dataset by maximizing the ELBO. We plot the train ELBO (left column), test 100-sample bound (middle column), and the variance of gradient estimator (right column) for the linear (top row) and nonlinear (bottom row) models. We plot the mean and one standard error based on 5 runs from different random initializations. Results on MNIST and Omniglot were qualitatively similar (Appendix Figure 6).

## 5.2 Training a Bernoulli VAE with ELBO

We now consider the much more challenging problem of training a VAE with Bernoulli latent variables, which is used as a gradient estimator benchmark for discrete latent variables. We evaluate the gradient estimators on three benchmark generative modeling datasets: MNIST, FashionMNIST

Table 1: Mean variational lower bounds and the standard error of the mean computed based on 5 runs from different random initializations. The best performing method (up to the standard error) for each task is in bold. To provide a computationally fair comparison between VIMCO $2K$-samples and DisARM $K$-pairs, we report the $2K$-sample bound for both, even though DisARM optimizes the $K$-sample bound. Results are for single stochastic layer models unless stated otherwise.

| | Train ELBO | | | |
|---|---|---|---|---|
| **Dynamic MNIST** | REINFORCE LOO | ARM | DisARM | RELAX |
| Linear | $-116.57 \pm 0.15$ | $-117.66 \pm 0.04$ | $\mathbf{-116.30 \pm 0.08}$ | $-115.93 \pm 0.15$ |
| Nonlinear | $\mathbf{-102.45 \pm 0.12}$ | $-107.32 \pm 0.28$ | $\mathbf{-102.56 \pm 0.19}$ | $-102.53 \pm 0.15$ |
| **Fashion MNIST** | | | | |
| Linear | $-256.33 \pm 0.14$ | $-256.80 \pm 0.16$ | $\mathbf{-255.97 \pm 0.07}$ | $-255.83 \pm 0.03$ |
| Nonlinear | $\mathbf{-237.66 \pm 0.11}$ | $-241.30 \pm 0.10$ | $\mathbf{-237.77 \pm 0.08}$ | $-238.23 \pm 0.17$ |
| **Omniglot** | | | | |
| Linear | $-121.66 \pm 0.10$ | $-122.45 \pm 0.10$ | $\mathbf{-121.15 \pm 0.12}$ | $-120.79 \pm 0.09$ |
| Nonlinear | $\mathbf{-115.26 \pm 0.15}$ | $-118.76 \pm 0.05$ | $\mathbf{-115.08 \pm 0.11}$ | $-116.56 \pm 0.15$ |
| 2-Layer Linear | $-116.81 \pm 0.08$ | $-117.74 \pm 0.14$ | $\mathbf{-116.38 \pm 0.10}$ | $-115.45 \pm 0.08$ |
| 3-Layer Linear | $-115.20 \pm 0.08$ | $-116.18 \pm 0.13$ | $\mathbf{-114.81 \pm 0.09}$ | $-113.83 \pm 0.06$ |
| 4-Layer Linear | $-114.83 \pm 0.13$ | $-116.01 \pm 0.14$ | $\mathbf{-114.09 \pm 0.09}$ | $-113.64 \pm 0.14$ |

| | Train multi-sample bound | | | |
|---|---|---|---|---|
| **Dynamic MNIST** | DisARM 1-pair | VIMCO 2-samples | DisARM 10-pairs | VIMCO 20-samples |
| Linear | $\mathbf{-114.06 \pm 0.13}$ | $-115.80 \pm 0.08$ | $\mathbf{-108.61 \pm 0.08}$ | $-109.40 \pm 0.07$ |
| Nonlinear | $\mathbf{-100.80 \pm 0.11}$ | $-101.14 \pm 0.10$ | $\mathbf{-93.89 \pm 0.06}$ | $-94.52 \pm 0.05$ |
| **Fashion MNIST** | | | | |
| Linear | $\mathbf{-254.15 \pm 0.09}$ | $-255.41 \pm 0.10$ | $\mathbf{-247.77 \pm 0.08}$ | $-249.60 \pm 0.11$ |
| Nonlinear | $-236.91 \pm 0.10$ | $\mathbf{-236.41 \pm 0.10}$ | $\mathbf{-231.34 \pm 0.06}$ | $-232.01 \pm 0.08$ |
| **Omniglot** | | | | |
| Linear | $\mathbf{-119.89 \pm 0.06}$ | $-121.66 \pm 0.08$ | $\mathbf{-116.70 \pm 0.03}$ | $-117.68 \pm 0.07$ |
| Nonlinear | $-114.45 \pm 0.06$ | $\mathbf{-114.18 \pm 0.07}$ | $\mathbf{-108.29 \pm 0.04}$ | $\mathbf{-108.37 \pm 0.05}$ |

and Omniglot. As our goal is optimization, we use dynamic binarization to avoid overfitting and we largely find that training performance mirrors test performance. We use the standard split into train, validation, and test sets. See Appendix D for further implementation details.

We use the same model architecture as Yin and Zhou (2019). Briefly, we considered linear and nonlinear models. The nonlinear model used fully connected neural networks with two hidden layers of 200 leaky ReLU units (Maas et al., 2013). Both models had a single stochastic layer of 200 Bernoulli latent variables. The models were trained with Adam (Kingma and Ba, 2015) using a learning rate $10^{-4}$ on mini-batches of 50 examples for $10^6$ steps.

During training, we measure the training ELBO, the 100-sample bound on the test set, and the variance of the gradient estimator for the inference network averaged over parameters[3] and plot the results in Figure 2 for FashionMNIST and Appendix Figure 6 for MNIST and Omniglot. We report the final training results in Table 1 and test results in Appendix Table 2. We find a substantial performance gap between ARM and REINFORCE LOO, DisARM, or RELAX across all measures and configurations. We compared our implementation of ARM with the open-source implementation provided by Yin and Zhou (2019) and find that it replicates their results. Yin and Zhou (2019) evaluate performance on the *statically binarized* MNIST dataset, which is well known for overfitting and substantial overfitting is observed in their results. In such a situation, a method that performs worse at optimization may lead to better generalization. Additionally, they report the variance of the gradient estimator w.r.t. logits of the latent variables instead, which explains the discrepancy in the variance plots. Unlike the inference network parameter gradients, the logit gradients have no special significance as they are backpropagated into the inference network rather than used to update

parameters directly. We use the same architecture across methods and implement the estimators in the same framework to ensure a fair comparison.

DisARM has reduced gradient estimator variance over REINFORCE LOO across all models and datasets. This translates to consistent improvements over REINFORCE LOO with linear models and comparable performance on the nonlinear models across all datasets. For linear networks, RELAX achieves lower gradient estimator variance and better performance. However, this does not hold for nonlinear networks. For nonlinear networks across three datasets, RELAX initially has lower variance gradients, but DisARM overtakes it as training proceeds. Furthermore, training the model on a P100 GPU was nearly twice as slow for RELAX, while ARM, DisARM and REINFORCE LOO trained at the same speed. This is consistent with previous findings (Yin and Zhou, 2019).

## 5.3    Training a Hierarchical Bernoulli VAE with ELBO

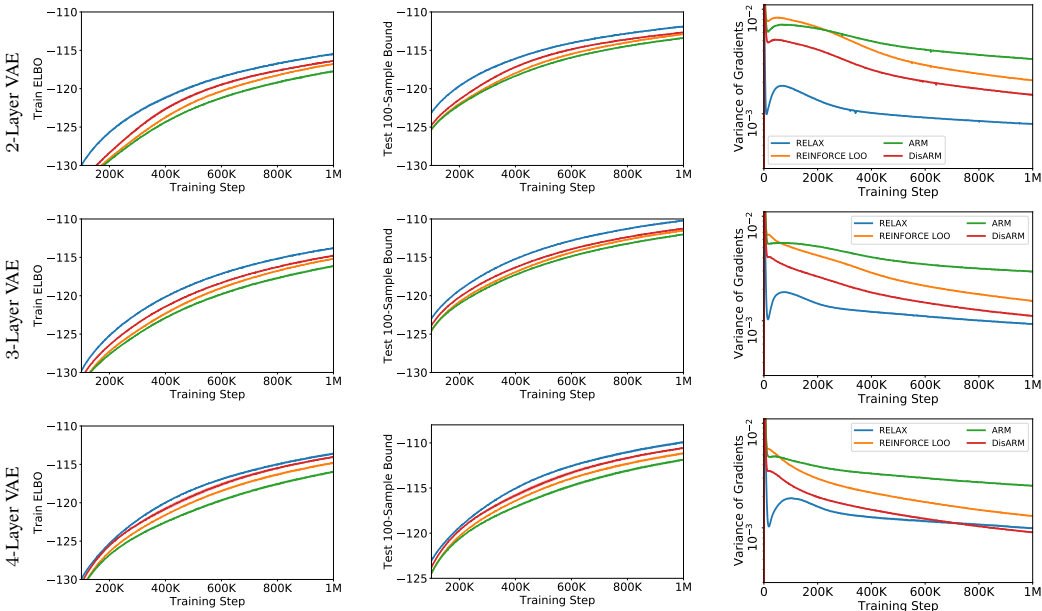

Figure 3: Training 2/3/4-layer Bernoulli VAE on Omniglot using DisARM, RELAX, REINFORCE LOO, and ARM. We report the ELBO on the training set (left), the 100-sample bound on the test set (middle), and the variance of the gradient estimator (right).

To compare the performance of the gradient estimators when scaling to hierarchical VAE models, we followed the techniques used in (Tucker et al., 2017; Grathwohl et al., 2018; Yin and Zhou, 2019) to extend DisARM to this setting (summarized in Appendix Algorithm 1). We evaluate Bernoulli VAE models with 2, 3 and 4 linear stochastic hidden layers on MNIST, Fashion-MNIST, and Omniglot datasets. Each linear stochastic hidden layer is of 200 units. We plot the results in Figure 3 for Omniglot and in Figure 7 for MNIST and FashionMNIST. We report the final training results in Table 1 for Omniglot, and full training and test results across all datasets in Appendix Table 3. We find that DisARM consistently outperforms ARM and REINFORCE-LOO. RELAX outperforms DisARM, however, the gap between two estimators diminishes for deeper hierarchies and training with DisARM is about twice as fast (wall clock time) as with RELAX.

## 5.4    Training a Bernoulli VAE with Multi-sample Bounds

To ensure a fair comparison on computational grounds, we compare the performance of models trained using DisARM with $K$ pairs of antithetic samples to models trained using VIMCO with $2K$ independent samples. For all of the performance results, we use the $2K$-sample bound, which favors VIMCO because this is precisely the objective it maximizes.

In order for a comparison of gradient estimator variances to be meaningful, the estimators must be unbiased estimates of the same gradient. So for the variance comparison, we compare DisARM with

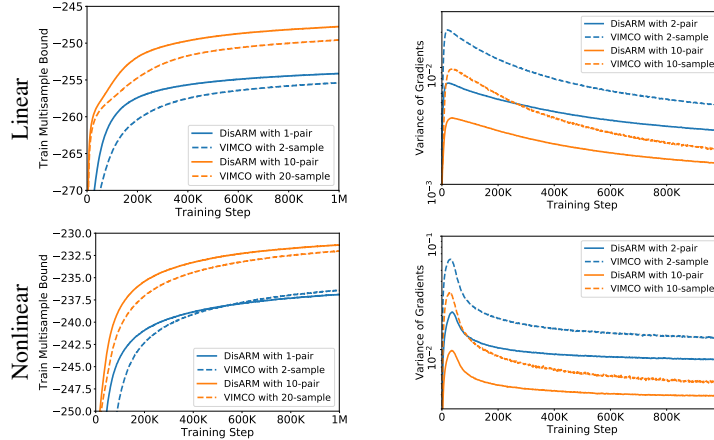

Figure 4: Training a Bernoulli VAE on FashionMNIST by maximizing the multi-sample variational bound with DisARM (solid line) and VIMCO (dashed line). We report the training multi-sample bound and the variance of the gradient estimators for the linear and nonlinear models. Test performance and results on MNIST and Omniglot were qualitatively similar (Appendix Figure 8).

$K$ pairs to averaging two independent VIMCO estimators with $K$ samples so that they use the same amount of computation. Furthermore, we compute the variance estimates along the same model trajectory (generated by VIMCO updates).

As shown in Figure 4, Table 1, Appendix Figure 8, and Appendix Table 4, DisARM consistently improves on VIMCO across different datasets, network settings, and number of samples/pairs.

## 6 Discussion

We have introduced DisARM, an unbiased, low-variance gradient estimator for Bernoulli random variables based on antithetic sampling. Our starting point was the ARM estimator (Yin and Zhou, 2019), which reparameterizes Bernoulli variables in terms of Logistic variables and estimates the REINFORCE gradient over the Logistic variables using antithetic sampling. Our key insight is that the ARM estimator involves unnecessary randomness because it operates on the augmenting Logistic variables instead of the original Bernoulli ones. In other words, ARM is competitive despite rather than because of the Logistic augmentation step, and its low variance is completely due to the use of antithetic sampling. We derive DisARM by integrating out the augmenting variables from ARM using a variance reduction technique known as conditioning. As a result, DisARM has lower variance than ARM and consistently outperforms it. Then, we extended DisARM to the multi-sample objective and showed that it outperformed the state-of-the-art method. Given DisARM's generality and simplicity, we expect it to be widely useful.

While relaxation-based estimators (e.g., REBAR and RELAX) can outperform DisARM in some cases, DisARM is always competitive and more generally applicable as it does not rely on a continuous relaxation. In the future, it would be interesting to investigate how to combine the strengths of DisARM with those of relaxation-based estimators in a single estimator. Finally, ARM has been extended to categorical variables (Yin et al., 2019), and in principle, the idea for DisARM can be extended to categorical variables. However, we do not yet know if the analytic integration can be done efficiently in this case.

## Broader Impacts

Gradient estimators for discrete latent variables have particular applicability to interpretable models and modeling natural systems with discrete variables. Discrete latent variables tend to be easier to interpret than continuous latent variables. While interpretable systems are typically viewed as a positive, they only give a partial view of a complex system. If they are not used with care and presented to the user properly, they may give the user a misplaced sense of trust. Providing simple and effective foundational tools enables non-experts to contribute, however, it also enables bad actors.

## Acknowledgments and Disclosure of Funding

We thank Chris J. Maddison and Michalis Titsias for helpful comments. We thank Mingzhang Yin for answering implementation questions about ARM.

## Footnotes

[1]"Baseline" can also refer to the scaling coefficient of the score function.

[2]Yin and Zhou (2019) show that ARM outperforms RELAX on this task, so we omit it.

[3]Estimated by approximating moments with an exponential moving average with decay rate 0.999.

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
