[Supplementary Material]

# A  DisARM Derivation

To finish the derivation of Eq. 6, we need to compute

$$\mathbb{E}_{q(z|b,\tilde{b})}\left[\nabla_\theta \log q_\theta(z)\right] = \mathbb{E}_{q(z|b,\tilde{b})}\left[1 - \frac{2\exp(-(z-\alpha_\theta))}{1+\exp(-(z-\alpha_\theta))}\right]\nabla_\theta\alpha_\theta \tag{10}$$

$$= \mathbb{E}_{q(u|b,\tilde{b})}\left[2u-1\right]\nabla_\theta\alpha_\theta = \left(2\mathbb{E}_{q(u|b,\tilde{b})}\left[u\right]-1\right)\nabla_\theta\alpha_\theta,$$

where we have used the change of variables $z = \log(u) - \log(1-u) + \alpha_\theta$, and thus $u = \sigma(z - \alpha_\theta)$. This is a common reparameterization of a Logistic variable in terms of a Uniform variable, so when $z \sim \text{Logistic}(\alpha_\theta, 1)$, then $u \sim \text{Uniform}(0,1)$. Thus, the joint distribution $q(u, b, \tilde{b})$ is generated by sampling $u \sim \text{Uniform}(0,1)$ and setting $b = \mathbb{1}_{z>0} = \mathbb{1}_{1-u<\sigma(\alpha_\theta)}$ and $\tilde{b} = \mathbb{1}_{\tilde{z}>0} = \mathbb{1}_{u<\sigma(\alpha_\theta)}$. Conditioning on $b, \tilde{b}$ imposes constraints on the value of $u$, hence $q(u|b, \tilde{b})$ is a truncated Uniform variable. To compute $\mathbb{E}_{q(u|b,\tilde{b})}\left[u\right]$, it suffices to enumerate the possibilities:

- $b = 0$, $\tilde{b} = 0$ implies $\sigma(\alpha_\theta) < u < \sigma(-\alpha_\theta)$, which is symmetric around $\sigma(0) = \frac{1}{2}$, so $\mathbb{E}_{q(u|b,\tilde{b})}\left[u\right] = \frac{1}{2}$.

- $b = 1$, $\tilde{b} = 1$ implies $\sigma(-\alpha_\theta) < u < \sigma(\alpha_\theta)$, which is symmetric around $\sigma(0) = \frac{1}{2}$, so $\mathbb{E}_{q(u|b,\tilde{b})}\left[u\right] = \frac{1}{2}$.

- $b = 0$, $\tilde{b} = 1$ implies $u < \min(\sigma(-\alpha_\theta), \sigma(\alpha_\theta)) = \sigma(-|\alpha_\theta|) = 1 - \sigma(|\alpha_\theta|)$. Thus,
$$\mathbb{E}_{q(u|b,\tilde{b})}\left[u\right] = \frac{1 - \sigma(|\alpha_\theta|)}{2}.$$

- $b = 1$, $\tilde{b} = 0$ implies $u > \max(\sigma(-\alpha_\theta), \sigma(\alpha_\theta)) = \sigma(|\alpha_\theta|)$. Thus,
$$\mathbb{E}_{q(u|b,\tilde{b})}\left[u\right] = \frac{1 + \sigma(|\alpha_\theta|)}{2}.$$

Combining the cases, we have that

$$2\mathbb{E}_{q(u|b,\tilde{b})}\left[u\right] - 1 = (-1)^{\tilde{b}}\mathbb{1}_{b\neq\tilde{b}}\sigma(|\alpha_\theta|).$$

# B  Interpolated Estimator

Depending on the properties of the function, antithetic samples can result in *higher* variance estimates compared to an estimator based on the same number of independent samples. This can be resolved by constructing an *interpolated estimator*.

Let $q_\theta^A(b, \tilde{b})$ be the antithetic Bernoulli distribution and $q_\theta^I(b, \tilde{b})$ by the independent Bernoulli distribution. Explicitly, when $p = \sigma(\alpha_\theta) < 0.5$, we have

$$q_\theta^A(b, \tilde{b}) = \begin{cases} 1 - 2p & b = \tilde{b} = 0, \\ 0 & b = \tilde{b} = 1, \\ p & \text{o.w.}, \end{cases}$$

and when $p \geq 0.5$

$$q_\theta^A(b, \tilde{b}) = \begin{cases} 0 & b = \tilde{b} = 0, \\ 2p - 1 & b = \tilde{b} = 1, \\ 1 - p & \text{o.w.} \end{cases}$$

Let $\beta \in [0,1]$ and define $q_\theta^\beta(b, \tilde{b}, i) = q^\beta(i)q_\theta(b, \tilde{b}|i)$ with $q^\beta(i) = \text{Bernoulli}(\beta)$ and $q_\theta(b, \tilde{b}|i) = iq_\theta^A(b, \tilde{b}) + (1-i)q_\theta^I(b, \tilde{b})$.

Then, the interpolated estimator is

$$g_{\text{Interpolated}}^\beta(b, \tilde{b}) = \mathbb{E}_{q_\theta^\beta(i|b,\tilde{b})}\left[ig_{\text{DisARM}}(b, \tilde{b}) + (1-i)g_{\text{LOO}}(b, \tilde{b})\right]$$

$$= q_\theta^\beta(i=1|b, \tilde{b})g_{\text{DisARM}}(b, \tilde{b}) + q_\theta^\beta(i=0|b, \tilde{b})g_{\text{LOO}}(b, \tilde{b}),$$

where explicitly

$$g_{\text{LOO}}(b, \tilde{b}) = \frac{1}{2} \left( (f(b) - f(\tilde{b}) \nabla_\theta \log q_\theta(b) + (f(\tilde{b}) - f(b)) \nabla_\theta \log q_\theta(\tilde{b}) \right).$$

Note that when $\beta = 0$, $g_{\text{Interpolated}}^\beta$ reduces to $g_{\text{LOO}}$ and when $\beta = 1$, it reduces to $g_{\text{DisARM}}$. To compute $q^\beta(i|b, \tilde{b})$, we use Bayes rule to rewrite it as

$$q^\beta(i = 1|b, \tilde{b}) = \frac{\beta q^A(b, \tilde{b})}{(1 - \beta)q^I(b, \tilde{b}) + \beta q^A(b, \tilde{b})},$$

in terms of known values.

From the definition, we have

$$\mathbb{E}_{q_\theta^\beta(b,\tilde{b})} \left[ g_{\text{Interpolated}}^\beta(b, \tilde{b}) \right] = \mathbb{E}_{q_\theta^\beta(b,\tilde{b},i)} \left[ i g_{\text{DisARM}}(b, \tilde{b}) + (1 - i) g_{\text{LOO}}(b, \tilde{b}) \right]$$

$$= \beta \mathbb{E}_{q_\theta^A(b,\tilde{b})} \left[ g_{\text{DisARM}}(b, \tilde{b}) \right] + (1 - \beta) \mathbb{E}_{q_\theta^I(b,\tilde{b})} \left[ g_{\text{LOO}}(b, \tilde{b}) \right],$$

so because $g_{\text{DisARM}}$ and $g_{\text{LOO}}$ are unbiased, $g_{\text{Interpolated}}^\beta$ is also unbiased. Because this estimator is unbiased for any choice of $\beta \in [0, 1]$, we can optimize $\beta$ to reduce variance as in (Ruiz et al. 2016; Tucker et al. 2017) and thus automatically choose the coupling which is favorable for the function under consideration.

## C Algorithm for Training Multi-layer Bernoulli VAE

For hierarchical VAEs, we use an inference network of the form $q_\theta(\mathbf{b}|x) = \prod_t q_\theta(\mathbf{b}_t|\mathbf{b}_{t-1}) = \prod_t \text{Bernoulli}(\mathbf{b}_t; \alpha_\theta(\mathbf{b}_{t-1}))$, where $\mathbf{b}_t$ is the set of binary latent variables for the $t$-th layer (with $\mathbf{b}_0 = x$ for convenience). The algorithm for computing the DisARM gradient estimator is summarized in Algorithm 1.

---

**Algorithm 1:** DisARM gradient estimator for a $T$-stochastic-hidden-layer binary network

---

**input** : A mini-batch $\boldsymbol{x}$ of data.

Initialize $g_\theta = 0$.
$\mathbf{b}_0 = \boldsymbol{x}$.
Sample $\boldsymbol{u}_{1:T} \sim \prod \text{Uniform}(0, 1)$
// Sample trunk.
**for** $t = 1{:}T$ **do**
$\quad | \quad \mathbf{b}_t = \mathbb{1}_{1-u_t < \sigma(\alpha_\theta(\mathbf{b}_{t-1}))}.$
**end**
**for** $t = 1{:}T$ **do**
$\quad$ // Antithetic sampling.
$\quad \tilde{\boldsymbol{b}}_t = \mathbb{1}_{u_t < \sigma(\alpha_\theta(\mathbf{b}_{t-1}))}.$
$\quad$ // Sample branch.
$\quad$ Sample $\tilde{\boldsymbol{b}}_{t+1:T} \sim q_\theta(\cdot|\tilde{\boldsymbol{b}}_t).$
$\quad f_\Delta = f(\boldsymbol{b}_{0:t-1}, \boldsymbol{b}_{t:T}) - f(\boldsymbol{b}_{0:t-1}, \tilde{\boldsymbol{b}}_{t:T}).$
$\quad g_\theta \mathrel{+}= \frac{1}{2} f_\Delta \sum_i \left( \left( (-1)^{\tilde{b}_{ti}} \mathbb{1}_{b_{ti} \neq \tilde{b}_{ti}} \sigma(|(\alpha_\theta(\boldsymbol{b}_{t-1}))_i|) \right) \nabla_\theta (\alpha_\theta(\boldsymbol{b}_{t-1}))_i \right).$
**end**
Return $g_\theta$.

---

## D Experimental Details

Input images to the networks were centered with the global mean of the training dataset. For the nonlinear network activations, we used leaky rectified linear units (LeakyReLU, Maas et al., 2013) activations with the negative slope coefficient of 0.3 as in (Yin and Zhou, 2019). The parameters of the inference and generation networks were optimized with Adam (Kingma and Ba, 2015) using learning rate $10^{-4}$. The logits for the prior distribution $p(b)$ were optimized using SGD with learning

rate $10^{-2}$ as in (Yin and Zhou, 2019). For RELAX, we initialize the trainable temperature and scaling factor of the control variate to $0.1$ and $1.0$, respectively. The learned control variate in RELAX was a single-hidden-layer neural network with 137 LeakyReLU units. The control variate parameters were also optimized with Adam using learning rate $10^{-4}$.

# E   Additional Experimental Results

Figure 5: Comparing gradient estimators for the toy problem (Section 5.1). We plot the trace of the estimated Bernoulli probability $\sigma(\phi)$, the estimated gradients, and the variance of the estimated gradients. The variance is measured based on $5000$ Monte-Carlo samples at each iteration.

In Appendix Figure 5, we compare gradient estimators for the toy problem Section 5.1, for which the exact gradient is

$$(1 - 2p_0)\sigma(\phi)(1 - \sigma(\phi)).$$

Trace plots for the estimated probability $\sigma(\phi)$ and the estimated gradients are similar for the three estimators, REINFORCE LOO, ARM and DisARM. However, DisARM exhibits lower variance than REINFORCE LOO and ARM, especially as the problem becomes harder with increasing $\phi$.

## Dynamic MNIST

## Omniglot

Figure 6: Training a Bernoulli VAE by maximizing the ELBO using DisARM (red), RELAX (blue), REINFORCE LOO (orange), and ARM (green). Both MNIST and Omniglot were dynamically binarized. We report the ELBO on training set (left column), the 100-sample bound on test set (middle column) and the variance of gradients (right column) for linear (top row) and nonlinear (bottom row) models. The mean and standard error (shaded area) are estimated given 5 runs from different random initializations.

Figure 7: Training 2/3/4-layer Bernoulli VAE on MNIST and FashionMNIST using DisARM, RELAX, REINFORCE LOO, and ARM. We report the ELBO on the training set (left), the 100-sample bound on the test set (middle), and the variance of the gradient estimator (right).

(a) Linear                                    (b) Noninear

Dynamic MNIST

(a) Linear                                    (b) Noninear

Fashion MNIST

(a) Linear                                    (b) Noninear

Omniglot

Figure 8: Training a Bernoulli VAE by maximizing the multi-sample variational bound with DisARM and VIMCO. We report the training and test multi-sample bound and the variance of the gradient estimators for the linear (a) and nonlinear (b) models. We evaluate the model on three datasets: MNIST, FashionMNIST and Omniglot, with dynamic binarization.

Table 2: Results for models trained by maximizing the ELBO. We report the mean and the standard error of the mean for the ELBO on the training set and of the 100-sample bound on the test set. The results we computed based on 5 runs from different random initializations and the standard error of the mean. The best performing method (up to the standard error) for each task is in bold.

| Train ELBO | | | | |
|---|---|---|---|---|
| **Dynamic MNIST** | REINFORCE LOO | ARM | DisARM | RELAX |
| Linear | $-116.57 \pm 0.15$ | $-117.66 \pm 0.04$ | $\mathbf{-116.30 \pm 0.08}$ | $-115.93 \pm 0.15$ |
| Nonlinear | $\mathbf{-102.45 \pm 0.12}$ | $-107.32 \pm 0.28$ | $\mathbf{-102.56 \pm 0.19}$ | $-102.53 \pm 0.15$ |
| **Fashion MNIST** | | | | |
| Linear | $-256.33 \pm 0.14$ | $-256.80 \pm 0.16$ | $\mathbf{-255.97 \pm 0.07}$ | $-255.83 \pm 0.03$ |
| Nonlinear | $\mathbf{-237.66 \pm 0.11}$ | $-241.30 \pm 0.10$ | $\mathbf{-237.77 \pm 0.08}$ | $-238.23 \pm 0.17$ |
| **Omniglot** | | | | |
| Linear | $-121.66 \pm 0.10$ | $-122.45 \pm 0.10$ | $\mathbf{-121.15 \pm 0.12}$ | $-120.79 \pm 0.09$ |
| Nonlinear | $\mathbf{-115.26 \pm 0.15}$ | $-118.76 \pm 0.05$ | $\mathbf{-115.08 \pm 0.11}$ | $-116.56 \pm 0.15$ |

| Test 100-sample bound | | | | |
|---|---|---|---|---|
| **Dynamic MNIST** | REINFORCE LOO | ARM | DisARM | RELAX |
| Linear | $\mathbf{-109.25 \pm 0.09}$ | $-109.70 \pm 0.05$ | $\mathbf{-109.13 \pm 0.04}$ | $-108.76 \pm 0.06$ |
| Nonlinear | $\mathbf{-97.41 \pm 0.09}$ | $-101.15 \pm 0.39$ | $\mathbf{-97.52 \pm 0.11}$ | $-97.76 \pm 0.11$ |
| **Fashion MNIST** | | | | |
| Linear | $-252.55 \pm 0.12$ | $-252.66 \pm 0.07$ | $\mathbf{-252.30 \pm 0.05}$ | $-252.13 \pm 0.06$ |
| Nonlinear | $\mathbf{-236.94 \pm 0.09}$ | $-239.37 \pm 0.15$ | $\mathbf{-237.02 \pm 0.07}$ | $-237.95 \pm 0.16$ |
| **Omniglot** | | | | |
| Linear | $-117.70 \pm 0.10$ | $-118.01 \pm 0.06$ | $\mathbf{-117.39 \pm 0.09}$ | $-117.10 \pm 0.08$ |
| Nonlinear | $\mathbf{-114.39 \pm 0.21}$ | $-116.56 \pm 0.07$ | $\mathbf{-114.26 \pm 0.14}$ | $-116.28 \pm 0.26$ |

Table 3: Results for training Bernoulli VAEs with 2/3/4 stochastic hidden layers. We report the mean and the standard error of the mean for the ELBO on training set and of the 100-sample bound on the test set. The results are computed based on 5 runs with different random initializations. The best performing methods (up to standard error) for each task is in bold.

|  | Train ELBO | | | |
| --- | --- | --- | --- | --- |
| **Dynamic MNIST** | REINFORCE LOO | ARM | DisARM | RELAX |
| 2-Layer | $-106.34 \pm 0.10$ | $-107.90 \pm 0.10$ | $\mathbf{-105.88 \pm 0.04}$ | $-105.48 \pm 0.04$ |
| 3-Layer | $-102.13 \pm 0.09$ | $-103.76 \pm 0.11$ | $\mathbf{-101.63 \pm 0.09}$ | $-101.22 \pm 0.09$ |
| 4-Layer | $-101.22 \pm 0.09$ | $-102.82 \pm 0.08$ | $\mathbf{-100.96 \pm 0.07}$ | $-99.86 \pm 0.07$ |
| **Fashion MNIST** | | | | |
| 2-Layer | $-244.67 \pm 0.16$ | $-245.76 \pm 0.11$ | $\mathbf{-244.04 \pm 0.06}$ | $-243.42 \pm 0.11$ |
| 3-Layer | $-239.88 \pm 0.03$ | $-241.21 \pm 0.12$ | $\mathbf{-239.64 \pm 0.06}$ | $-239.41 \pm 0.07$ |
| 4-Layer | $-238.86 \pm 0.09$ | $-239.99 \pm 0.04$ | $\mathbf{-238.49 \pm 0.08}$ | $-238.23 \pm 0.08$ |
| **Omniglot** | | | | |
| 2-Layer | $-116.81 \pm 0.08$ | $-117.74 \pm 0.14$ | $\mathbf{-116.38 \pm 0.10}$ | $-115.45 \pm 0.08$ |
| 3-Layer | $-115.20 \pm 0.08$ | $-116.18 \pm 0.13$ | $\mathbf{-114.81 \pm 0.09}$ | $-113.83 \pm 0.06$ |
| 4-Layer | $-114.83 \pm 0.13$ | $-116.01 \pm 0.14$ | $\mathbf{-114.09 \pm 0.09}$ | $-113.64 \pm 0.14$ |
|  | Test 100-sample bound | | | |
| **Dynamic MNIST** | REINFORCE LOO | ARM | DisARM | RELAX |
| 2-Layer | $-99.45 \pm 0.07$ | $-100.31 \pm 0.07$ | $\mathbf{-99.12 \pm 0.05}$ | $-98.65 \pm 0.03$ |
| 3-Layer | $-95.40 \pm 0.05$ | $-96.47 \pm 0.07$ | $\mathbf{-95.08 \pm 0.04}$ | $-94.53 \pm 0.06$ |
| 4-Layer | $-94.72 \pm 0.06$ | $-95.84 \pm 0.09$ | $\mathbf{-94.60 \pm 0.04}$ | $-93.34 \pm 0.04$ |
| **Fashion MNIST** | | | | |
| 2-Layer | $-241.98 \pm 0.14$ | $-242.58 \pm 0.11$ | $\mathbf{-241.42 \pm 0.05}$ | $-240.84 \pm 0.08$ |
| 3-Layer | $-237.80 \pm 0.06$ | $-238.59 \pm 0.13$ | $\mathbf{-237.59 \pm 0.09}$ | $-237.32 \pm 0.08$ |
| 4-Layer | $-237.09 \pm 0.07$ | $-237.72 \pm 0.05$ | $\mathbf{-236.78 \pm 0.09}$ | $-236.43 \pm 0.10$ |
| **Omniglot** | | | | |
| 2-Layer | $-112.92 \pm 0.04$ | $-113.39 \pm 0.10$ | $\mathbf{-112.64 \pm 0.06}$ | $-111.87 \pm 0.09$ |
| 3-Layer | $-111.52 \pm 0.07$ | $-112.01 \pm 0.09$ | $\mathbf{-111.25 \pm 0.08}$ | $-110.22 \pm 0.06$ |
| 4-Layer | $-111.16 \pm 0.11$ | $-111.87 \pm 0.11$ | $\mathbf{-110.58 \pm 0.08}$ | $-109.95 \pm 0.12$ |

Table 4: Train and test variational lower bounds for models trained using the multi-sample objective. We report the mean and the standard error of the mean computed based on 5 runs from different random initializations. The best performing method (up to the standard error) for each task is in bold. To provide a computationally fair comparison between VIMCO $2K$-samples and DisARM $K$-pairs, we report the $2K$-sample bound for both, even though DisARM optimizes the $K$-sample bound.

| Train multi-sample bound | | | | |
|---|---|---|---|---|
| **Dynamic MNIST** | DisARM 1-pair | VIMCO 2-samples | DisARM 10-pairs | VIMCO 20-samples |
| Linear | $\mathbf{-114.06 \pm 0.13}$ | $-115.80 \pm 0.08$ | $\mathbf{-108.61 \pm 0.08}$ | $-109.40 \pm 0.07$ |
| Nonlinear | $\mathbf{-100.80 \pm 0.11}$ | $-101.14 \pm 0.10$ | $\mathbf{-93.89 \pm 0.06}$ | $-94.52 \pm 0.05$ |
| **Fashion MNIST** | | | | |
| Linear | $\mathbf{-254.15 \pm 0.09}$ | $-255.41 \pm 0.10$ | $\mathbf{-247.77 \pm 0.08}$ | $-249.60 \pm 0.11$ |
| Nonlinear | $-236.91 \pm 0.10$ | $\mathbf{-236.41 \pm 0.10}$ | $\mathbf{-231.34 \pm 0.06}$ | $-232.01 \pm 0.08$ |
| **Omniglot** | | | | |
| Linear | $\mathbf{-119.89 \pm 0.06}$ | $-121.66 \pm 0.08$ | $\mathbf{-116.70 \pm 0.03}$ | $-117.68 \pm 0.07$ |
| Nonlinear | $-114.45 \pm 0.06$ | $\mathbf{-114.18 \pm 0.07}$ | $\mathbf{-108.29 \pm 0.04}$ | $\mathbf{-108.37 \pm 0.05}$ |

| Test multi-sample bound | | | | |
|---|---|---|---|---|
| **Dynamic MNIST** | DisARM 1-pair | VIMCO 2-samples | DisARM 10-pairs | VIMCO 20-samples |
| Linear | $\mathbf{-113.63 \pm 0.13}$ | $-115.31 \pm 0.07$ | $\mathbf{-108.18 \pm 0.08}$ | $-108.97 \pm 0.08$ |
| Nonlinear | $\mathbf{-102.03 \pm 0.10}$ | $\mathbf{-102.15 \pm 0.11}$ | $\mathbf{-94.78 \pm 0.07}$ | $-95.34 \pm 0.06$ |
| **Fashion MNIST** | | | | |
| Linear | $\mathbf{-256.14 \pm 0.10}$ | $-257.35 \pm 0.12$ | $\mathbf{-249.71 \pm 0.10}$ | $-251.52 \pm 0.13$ |
| Nonlinear | $-239.53 \pm 0.10$ | $\mathbf{-238.99 \pm 0.11}$ | $\mathbf{-233.82 \pm 0.08}$ | $-234.47 \pm 0.09$ |
| **Omniglot** | | | | |
| Linear | $\mathbf{-120.23 \pm 0.07}$ | $-121.99 \pm 0.08$ | $\mathbf{-117.29 \pm 0.04}$ | $-118.29 \pm 0.07$ |
| Nonlinear | $-118.96 \pm 0.07$ | $\mathbf{-118.36 \pm 0.11}$ | $\mathbf{-112.43 \pm 0.07}$ | $\mathbf{-112.42 \pm 0.07}$ |