[Reviews · NeurIPS 2020]

Review 1

Summary and Contributions: This work proposes an unbiased and low-variance gradient estimator DisARM for binary latent variables, which exploits antithetic sampling. Based on a recent estimator ARM, DisARM improves it by integrating out the continuous augmentation of ARM for substantial variance reduction. In addition, the authors extend DisARM for multi-sample variational bounds. Through extensive experiments, it shows that DisARM consistently improves ARM and achieves state-of-the-art.

Strengths: 1) This work inspects the recent gradient estimator ARM and explains why the variance reduction in ARM works. This not only reveals the potential issues when using continuous augmentation in terms of discrete latent variables, but also shows how antithetic sampling can greatly help reduce variance. 2) It improves ARM by integrating out the continuous augmentation using novel techniques and extends the proposed estimator to multi-sample bounds, which is highly competitive with VIMCO. 3) This paper also emphasizes a strong baseline REINFORCE with leave-one-out control variates, which was largely overlooked in previous works. This is beneficial for further research in this area.

Weaknesses: The main weakness lies in that this work does not address the setting with multi-layer latent variables, as mentioned in the ARM paper. Adding an additional section for this setting could provide more direct comparisons with ARM.

Correctness: mostly correct

Clarity: yes

Relation to Prior Work: nil

Reproducibility: Yes

Additional Feedback:


Review 2

Summary and Contributions: This paper proposes a new estimator for the derivative with respect to parameters of a discrete distribution. The authors show that a previous estimator (ARM) can be decomposed into two parts: a continuous augmentation and antithetic sampling. The authors show that using the antithetic sampling alone is sufficient and the continuous augmentation only increases variance. The experiments show that the new estimator has competitive performance compared to other recent estimators, while enjoying the benefit of simplicity.

Strengths: This paper makes a nice contribution to an important problem. The proposed estimator (DisARM) is simple to understand / implement, while has good theoretical justification as the authors show that the variance is strictly no worse than a recent method (ARM). The experiments are convincing. A large number of recent baselines are compared (ARM, RELAX, VIMCO, REINFORCE LOO). Even though the proposed approach is not the best in all benchmarks, it is sufficiently competitive and do not require additional learning (as in RELAX).

Weaknesses: I think the paper is quite complete without additional experiments, but I think there are several experiments that would strengthen the story. A somewhat interesting comparison is with biased estimators such as reparameterized Gumbel, which are often shown to perform well despite the bias. It would also be interesting to see how the method does in applications such as model based RL / actor-critic methods, where discretization is often unavoidable. For a VAE there are many ways to avoid backpropagating through discrete variables, such as relaxation or use of mixture distributions, so an application to discrete VAE faces competition from many different types of methods.

Correctness: I did not spot any major issues in the derivation. The experiment methods are clearly explained, and baselines are (claimed to be) consistent with the original implementation by the original authors, which is great.

Clarity: The paper is mostly clear. 2.1 can be confusing to someone who has not read the ARM paper or familiar with antithetic sampling, and it is best to be self contained. For example, the meaning of z, \tilde{z}, \alpha_\theta are not motivated and explained in words, and the antithetic sampling procedure are only explained in the footnote.

Relation to Prior Work: I find the discussion sufficient.

Reproducibility: Yes

Additional Feedback:


Review 3

Summary and Contributions: In one sentence, this paper uses the Rao-Blackwellization technique to reduce the gradient variance of the ARM estimator. In ARM, the gradient of a stochastic optimization problem with respect to the parameters of a binary distribution is estimated by first, augmenting the binary space to a continuous space and then by defining antithetical sampling in the continuous space. This paper argues that additional gradient noise is introduced in the augmentation space and it presents a simple approach to remove the noise by marginalizing out the augmented continuous variables. 

Strengths: * Novelty: In the past few years, a variety of approaches has been proposed for gradient estimation over binary variables in stochastic optimization problems. While the common approach is to use continuous relaxation (thanks to their low-variance but biased estimation), in certain scenarios one cannot easily apply continuous relaxation (for example in reinforcement learning problems where action space is discrete and reward is non-differentiable). In such cases, RELAX and ARM could be applied as they are unbiased and they don't rely on continuous relaxation of the target function. This paper discusses the additional variance introduced in the augmentation step of ARM, and it presents an approach to reduce it. The final outcome of the proposed estimators is that it uses antithetic sampling to lower the gradient variance similar to ARM but it doesn't introduce additional noise due to the augmentation. * Theoretical grounding: The authors do an excellent job of deriving the gradient estimator. Although the derivations seem complex, the proposed method is theoretically well-grounded. * empirical evaluation: The method is examined on toy examples as well as training VAEs and IWAEs. The method has been shown to be stronger than ARM and VIMCO, similar to REINFORCE LOO. However, RELAX seems to perform better on some VAE cases.

Weaknesses: * Hierarchical models: It is common to use hierarchical encoders for VAEs in the form of q(b_1, b_2, ..., b_n) = q(b_1) q(b_2 | b_1) ... q(b_n | b_{<n}) where b_1, b_2, ... b_n are groups of binary latent variables. Many existing gradient estimators such as REBAR and RELAX do not scale well to hierarchical models. It is not clear if the proposed model can be applied to hierarchical models. * Comparison against RELAX: In several cases, it seems that RELAX outperforms the proposed model. This might be because RELAX uses continuous relaxation for variance reduction without requiring the objective to be differentiable. This paper claims in several locations (for example in the discussion) that the proposed method is more generic than RELAX. But in reality, I believe that both RELAX and the proposed method make similar assumptions on the problem. It would great if the authors could clarify the differences and state why the proposed method is more generic. In my opinion, even if RELAX outperforms the proposed method in some cases, this paper approaches the problem of gradient estimation for binary variables from a totally different perspective, which makes it very valuable for the community.  * Experiments on cases where continuous relaxation is unavailable: The problem with examining the method on VAEs is that the objective in VAEs can be easily relaxed and differentiable continuous relations approaches often outperform REINFORCE-based estimators. It would be very valuable if authors could examine the proposed method on a problem (such as reinforcement learning) in which the objective cannot be relaxed easily. For example, the original RELAX paper presented results for several RL problems.

Correctness: Yes, it is correct. The experiments seem good. Some additional experiments however could make the submission stronger (discussed above).

Clarity: The paper is well written. I really enjoyed reading it.

Relation to Prior Work: The prior work is discussed properly in my opinion. 

Reproducibility: Yes

Additional Feedback: *** After rebuttal *** After reading the author's response, I am going to keep my original. I believe this is a good submission and I highly encourage the authors to include their experiments on the hierarchical models in the final paper as well. Please include a description of how the proposed estimator can be applied to a hierarchical model. ************************


Review 4

Summary and Contributions: The paper presents a new method to estimate the gradient of an expectation with respect to the parameters of the integrating distribution, for the case where this distribution consists on independent binary variables. Getting low variance gradient estimators in this setting is challenging. This work builds on the method "ARM" (Yin and Zhou (2019)), which parameterizes the discrete variables as a deterministic transformation of some continuous variables, and then estimates the gradient using antithetic sampling. The authors first present a simple analysis that shows that the continuous augmentation used by ARM actually leads to an increase in the variance of the estimator, and that the method works well thanks to the use of antithetic sampling. With this in mind, the authors propose a way to analytically integrate out the continuous (augmentation) variables, which leads to an estimator with provable lower variance than the one obtained using ARM. Simply put, the method proposed retains the (potential) benefits of antithetic sampling without paying the price of using the augmentation with continuous random variables.

Strengths: Novelty: The estimator proposed, while closely related to previous work, is novel, and has theoretical advantages over the previous approach on which it builds, ARM. Significance and relevance: I believe this represents a relevant contribution to the community. Inference with discrete variables is an active area of research and this method appears to achieve state of the art performance on representative tasks.

Weaknesses: I identify two limitations with the method: - It is only applicable to binary independent random variables. Potential extensions to other cases (categorical and/or non-independent) would be very interesting. - It is based on the belief that antithetic sampling will actually be useful. While it has been proved antithetic samples have perfectly negative correlated score functions, the gradient contains the score function and an extra term ('f' in the paper). This extra term, depending on its properties, may render antithetic sampling useless or even unfavorable (it can be shown that the proposed method has lower variance than ARM, but, up to my understanding, there's no guarantee that it will have lower variance than the typical score function estimator. This will depend on this extra term/function.) Despite this, empirical evidence seems to suggest that this method is quite useful in practice, at least for the VAE models on which it was tested.

Correctness: The theoretical claims seem to be correct, and the empirical methodology too.

Clarity: Yes.

Relation to Prior Work: Yes.

Reproducibility: Yes

Additional Feedback: *** After rebuttal *** After reading the author's response, I choose to update my score from 6 to 7. The authors addressed my main concern about the use of antithetic samples. While it is not ideal that the method may reduce to the reinforce estimator when antithetic sampling is not useful, I think this approach can be combined with other variance reduction techniques that may help in such scenarios. *** ***

[Author Response · NeurIPS 2020]

We thank the reviewers for their suggestions. We have revised the paper for clarity and added experiments on hierarchical
models requested by reviewers.

**R2, R4: Application to hierarchical models.** Closely following the techniques used in (Tucker et al. 2017; Grathwohl
et al. 2018; Yin and Zhou 2019), we extend DisARM to hierarchical models. We evaluate $2/3/4$-layer linear models
on MNIST, Fashion-MNIST, and Omniglot (Figure 1 shows the Omniglot results) and find that DisARM consistently
outperforms ARM and REINFORCE-LOO. RELAX outperforms DisARM, however, the gap between two estimators
diminishes for deeper hierarchies and training with DisARM is about twice as fast (wall clock time) as with RELAX.

**R3: Section 2.1 clarity.** We have clarified this section by moving definitions into the main text and providing motivation
and intuition for the choices.

**R4: Differences with RELAX.** RELAX requires gradients from a (learned) surrogate function. While in principle
RELAX is generic because the surrogate can be learned from scratch, the strong performance previously reported
(and in this paper) relies on a continuous relaxation of the discrete function and only learns a small deviation from
this hard-coded relaxation. Moreover, for discrete VAEs, using the continuous relaxation has the same computational
cost as working with the discrete model, but in other problems it can be much slower. For example, with conditional
computation, the continuous relaxation requires evaluating the entire model, while pure discrete approaches, such as
DisARM, evaluate only the parts of the model selected by the discrete gates.

**R5: Extension to categorical variables.** Yes, in principle the idea for DisARM can be extended to categorical
variables. The authors of ARM released an extension ARSM (Yin et al. 2019) for categorical variables and the same
idea of analytic integration can be adapted to ARSM to reduce variance. However, we do not yet know if the analytic
integration can be done efficiently in this case.

**R3, R4: Application to RL.** We agree that this would be interesting, and we expect to see similar improvements
compared to ARM. However, this would require extending DisARM to the categorical case. Due to this and the
complexity of proper evaluation in RL, we feel applications to RL are beyond the scope of this paper.

**R5: Usefulness of antithetic samples.** The reviewer is correct that depending on the properties of the function,
antithetic samples can result in higher variance compared to the same number of independent samples. This can be
resolved by constructing an *interpolated estimator*. Briefly, we define a coupling between Bernoulli variables that is
parameterized by $\alpha \in [0, 1]$ that smoothly interpolates between independent and antithetic samples. Furthermore, we
construct an unbiased estimator parameterized by $\alpha$ for these coupled variables and such that $\alpha = 0$ corresponds to
REINFORCE LOO and $\alpha = 1$ corresponds to DisARM. Because this estimator is unbiased for any choice of $\alpha \in [0, 1]$,
we can optimize $\alpha$ to reduce variance as in (Ruiz et al. 2016; Tucker et al. 2017) and thus automatically choose the
coupling which is favorable for the function under consideration. We have added an appendix section describing this
construction and now mention it in the main text. In preliminary experiments, we did not find significant improvements
on the datasets we evaluated.

Figure 1: Training 2/3/4-layer Bernoulli VAE on Omniglot using DisARM, RELAX, REINFORCE LOO, and ARM. We report the
ELBO on the training set (left), the 100-sample bound on the test set (middle), and the variance of the gradient estimator (right).

[Meta-Review · NeurIPS 2020]

The reviewers agree that this submission represents an important contribution to the field. Please be sure to carefully review and address the concerns of all reviewers in the revision.